# Unraveling Racial Disparities in Papillary Thyroid Cancer: A Comparative Bulk RNA-Sequencing Gene Expression Analysis

**DOI:** 10.3390/curroncol32060315

**Published:** 2025-05-29

**Authors:** Luiza Barseghyan, Samuel Chan, Celina R. Yamauchi, Andrea Shields, Mia C. Perez, Alfred A. Simental, Salma Khan

**Affiliations:** 1Center for Health Disparities, Loma Linda University, Loma Linda, CA 92350, USA; lbarseghyan@students.llu.edu (L.B.); samuelchan@students.llu.edu (S.C.); ryamauchi@llu.edu (C.R.Y.); 2Department of Pathology and Human Anatomy, Loma Linda University, Loma Linda, CA 92350, USA; ashields@llu.edu (A.S.); miperez@llu.edu (M.C.P.); 3Department of Otolaryngology-Head & Neck Surgery, Loma Linda University, Loma Linda, CA 92350, USA; asimenta@llu.edu; 4Center for Health Disparities & Molecular Medicine, 11085 Campus Street, Loma Linda, CA 92350, USA

**Keywords:** papillary thyroid cancer, ethnicity, RNA-sequencing, pathway analysis, TCGA, overall survival

## Abstract

Papillary thyroid cancer (PTC) is the most common thyroid malignancy, with significant racial/ethnic disparities in incidence and survival. Asians have the highest incidence, and recurrence, while African Americans experience the lowest survival rates, suggesting contributions from genetic, environmental, and healthcare-related factors. While socioeconomic disparities play a role, emerging evidence highlights genetic and molecular mechanisms underlying these differences. This study examines differentially expressed genes (DEGs) to identify potential molecular drivers of PTC disparities. Bulk RNA-sequencing (RNA-seq) data from 20 PTC tumors (5 White, 5 African American, 5 Hispanic, and 5 Asian) were analyzed using the UseGalaxy platform. Preprocessing included quality control, adapter trimming, and genome alignment. Differential expression analysis identified genes with *p* < 0.01 and fold change ≥ 2.5. Volcano plots visualized significant DEGs. Gene Set Enrichment Analysis (GSEA) via eVITTA identified enriched pathways. TCGA data analysis validated racial/ethnic differences in gene expression. Ethnic groups exhibited distinct gene expression profiles. GSEA revealed differences in cell proliferation, immune regulation, and thyroid hormone metabolism. African Americans showed immune suppression and reduced tumor suppressor activity, while Asians exhibited enriched cell cycle and DNA repair pathways. Significant differences were confirmed in some of the genes in TCGA data analysis. This study identifies genetic factors contributing to racial disparities in PTC, emphasizing the need for further validation in larger cohorts and functional studies. Understanding these molecular differences may inform personalized treatment strategies and improve PTC outcomes across diverse populations.

## 1. Introduction

Papillary thyroid cancer (PTC) is the most common form of thyroid cancer, accounting for roughly 80–85% of all thyroid malignancies [1]. Originating in the follicular cells of the thyroid gland, PTC is distinguished by its unique papillary histological features. Typically, it presents as a slow-growing tumor with an excellent prognosis when detected and treated early, boasting a 10-year survival rate exceeding 90% for most patients [2].

Over the past few decades, PTC incidence has steadily increased worldwide, partly due to improved diagnostic methods, such as high-resolution ultrasound and fine-needle aspiration biopsy, which have facilitated earlier detection of small, often asymptomatic tumors [3]. However, this rise in incidence is not solely attributed to enhanced detection techniques, suggesting that environmental factors and lifestyle changes may also play a role [4].

Patients with PTC commonly present with a painless thyroid nodule, frequently discovered incidentally during routine examinations or imaging studies for unrelated concerns [2]. Some patients may experience symptoms such as a visible neck lump, difficulty swallowing, hoarseness, or neck pain. Lymph node metastases are also common, found in approximately 30–50% of patients at diagnosis [5].

Genetically, the most frequent mutation in PTC is the *BRAF V600E* mutation, present in around 40–45% of cases, which leads to the constitutive activation of the MAPK signaling pathway and promotes cell growth and survival [6]. Other notable genetic alterations include *RET/PTC* rearrangements and *RAS* gene mutations [7,8].

PTC treatment typically involves surgical resection, often followed by radioactive iodine therapy to eliminate any remaining thyroid tissue or metastases [2]. The extent of surgery (total thyroidectomy versus lobectomy) and the decision to use radioactive iodine depend on factors like tumor size, lymph node involvement, and patient age. Long-term management includes thyroid hormone replacement therapy and routine monitoring for recurrence [9].

Despite the generally favorable prognosis of PTC, significant disparities exist in incidence and survival rates across different ethnic groups. Asians, for example, show the highest incidence and recurrence rates, while African Americans experience the lowest survival rates [4]. These disparities arise from a complex interplay of socioeconomic, environmental, and biological factors, though the genetic contributions remain poorly understood [4,10,11].

Advances in high-throughput sequencing technologies now enable comprehensive analysis of gene expression profiles in cancer tissues [12,13]. These methods hold promises for identifying ethnic-specific genetic alterations that may contribute to PTC outcome disparities. Uncovering these genetic differences could lead to more personalized and effective treatment strategies for diverse populations [14]. Our lab has previously demonstrated differential expressions of miRNAs and vitamin D binding proteins between White and Asian (Filipino American) patients [15,16]. Building on this work, in this study, we investigated genetic factors underlying ethnic disparities in PTC through Next Generation Sequencing (NGS) and differential gene expression analysis. By comparing gene expression profiles among Non-Hispanic White, African, Hispanic, and Asian American patients with PTC, we aim to identify ethnic-specific genetic signatures and biological pathways that may impact disease progression and treatment response. This research could deepen our understanding of the molecular basis of PTC across different ethnic groups, ultimately improving risk stratification and enabling targeted therapies for patients from diverse backgrounds.

## 2. Materials and Methods

### 2.1. Tumor Sample Selection

The samples were obtained from the Department of Pathology and Human Anatomy at Loma Linda University Medical Center (LLUMC) with IRB approval for the use of discarded materials. We collected 20 deidentified, discarded thyroid cancer samples from both male and female patients, ensuring equal representation of Non-Hispanic White, African American, Hispanic, and Asian American individuals with classical papillary thyroid carcinoma (PTC), as detailed in the Appendix A (ethnicity was recorded based on self-identification).

All pathological diagnoses were confirmed by expert pathologists using established histopathological criteria and routine hematoxylin and eosin (H&E) staining. To maintain genetic uniformity and tissue specificity, we included PTC subtypes (Classical) from all represented ethnic groups. This study was conducted in accordance with the guidelines of the Declaration of Helsinki, the Belmont Report, and the U.S. Common Rule. As per 45 CFR 46.101(b)(4), this study involved retrospective, deidentified clinical data and was deemed exempt from IRB review by LLU IRB. Consequently, patient consent was not required.

### 2.2. Total RNA Isolation from FFPE Samples

Tissue blocks were sent to CARIS Life Sciences (Irving, TX, USA) for Bulk RNA sequencing. Microdissected cancer samples were obtained after being postmarked by pathologists. The Caris Assure workflow prepares both DNA libraries and sequences simultaneously in a single sequencing run using a custom hybridization/capture methodology. For total RNA extraction from formalin-fixed paraffin-embedded (FFPE) papillary thyroid cancer (PTC) samples, we utilized the AllPrep DNA/RNA FFPE Kit (QIAGEN, Valencia, CA, USA). To ensure uniformity, only papillary thyroid cancer tissues were included. Each extraction began with the deparaffinization of thyroid slides cut to 5 microns using Xylene. The FFPE samples were then incubated in a Proteinase K lysis buffer to digest the tissue, which led to the release of RNA and the precipitation of DNA. Following centrifugation, the supernatant containing RNA and the pellet containing DNA were processed separately for their purification. Additional incubation steps were carried out to partially reverse crosslinking. RNA was purified using an RNeasy MinElute spin column, with an on-column DNase treatment to eliminate any remaining DNA. The quality and quantity of the extracted RNA were assessed using Nanodrop (NanoDrop Technologies, Waltham, MA, USA), and samples with an OD 260/280 ratio of less than 1.8 were excluded from further analysis.

### 2.3. RNA Sequencing

RNA-seq raw data were obtained from 19 out of 20 PTC samples (one sample did not show any data), comprising 5 Non-Hispanic Whites, 5 African Americans, 5 Hispanics, and 4 Asian Americans/Pacific Islanders. RNA extraction, library preparation, and sequencing were performed according to standard protocols from CARIS Life Sciences.

### 2.4. Data Analysis

#### 2.4.1. Data Acquisition and Quality Control

The data generated in this study are available within the article and its Appendix A. RNA sequencing of the raw data was obtained in BAM format, representing aligned reads for samples from different ethnic groups, including White and other ethnicities. Initial quality control of raw sequencing data was performed using FastQC (https://usegalaxy.org/?tool_id=toolshed.g2.bx.psu.edu%2Frepos%2Fdevteam%2Ffastqc%2Ffastqc%2F0.71, accessed on 11 August 2024) to ensure high-quality input for subsequent analyses.

#### 2.4.2. Alignment and Feature Counting

Salmon was run on fastqs to quantify expression at the transcript level using Ensembl annotations from the GRCh38 reference transcriptome. Read counts and TPMs for each transcript corresponding to a given gene were then summed to generate gene-level counts and TPMs. Gene-level read counts were generated using the featureCounts tool, resulting in a count matrix with genes as rows and samples as columns.

#### 2.4.3. Differential Expression Analysis

Based on the feature count files, we performed differential expression analyses through pairwise comparisons between White and other ethnic groups using DESeq2.

#### 2.4.4. DESeq2 Analysis

The count matrix was imported into DESeq2 for differential expression analysis. DESeq2 was employed to normalize the data, accounting for differences in sequencing depth across samples. A negative binomial generalized linear model was fitted to the data to test for differential expression between the ethnic groups. Pairwise comparisons were performed between White and other ethnic groups. Genes with an adjusted *p*-value < 0.05 were considered significantly differentially expressed.

#### 2.4.5. Visualization and Functional Analysis

Differentially expressed genes were visualized using Volcano plots within the Galaxy platform. The differentially expressed files were uploaded directly to the Galaxy history panel for processing. The input data consisted of a tab-delimited file containing columns for raw *p*-values, adjusted *p*-values (FDR), log fold change, and gene identifiers. An adjusted *p*-value of <0.05 was used.

#### 2.4.6. Gene Set Enrichment Analysis

Gene Set Enrichment Analysis (GSEA) was conducted using the eVITTA platform to further explore the differential expression results obtained from DESeq2. The analysis began with the generation of ranked gene lists from the DESeq2 output, where genes were ranked based on their log2 fold changes from the differential expression analysis comparing White and other ethnic groups. These ranked lists were then uploaded to eVITTA, an integrative tool that facilitates complex genomic data analyses and visualization.

For GSEA, the ranked gene lists were analyzed against predefined gene sets from the Hallmark collection, which encompasses a wide range of biological pathways. eVITTA automatically performed the enrichment analysis, calculating enrichment scores for each gene set and assessing their significance through permutation testing. The results were visualized using enrichment plots and tables, highlighting pathways with significant enrichment (adjusted *p*-value < 0.05), thereby identifying key biological processes associated with the observed gene expression changes.

#### 2.4.7. TCGA Data Validation

To ensure the robustness and reliability of our findings, validation was conducted using The Cancer Genome Atlas (TCGA) dataset. This large-scale genomic resource provides comprehensive multiomics data across various cancer types, enabling cross-validation of gene expression patterns identified in our study.

Our validation process began with the retrieval of relevant TCGA RNA-seq data, focusing on tumor samples stratified by ethnicity. Standardized preprocessing steps, including normalization and batch effect correction, were applied to ensure comparability across samples. Differential expression analysis was then conducted using DESeq2 to identify genes exhibiting consistent patterns with our primary dataset. The results were assessed in concordance with our initial findings, reinforcing the biological significance of the identified gene expression changes. Pathways and gene signatures showing consistent enrichment in both datasets provided further validation of their relevance to disease mechanisms.

## 3. Results

### 3.1. Gene Expression Profiles: Comparison Between White and Asian Papillary Thyroid Cancer Tissue Samples

The comparison between White and Asian PTC samples revealed that 2384 genes were differentially expressed (*p* < 0.05). The top 10 differentially expressed genes by adjusted *p*-value in White vs. Asian PTC samples are shown in the Volcano plot (Figure 1a). The downregulated genes were *MIR205HG*, superoxide dismutase (*SOD3*), hypoxia upregulated protein 1 (*HYOU1*), solute carrier family 16 member 9 (*SLC16A9*), potassium voltage-gated channel (*KCNQ1*), Nicotinamide Mononucleotide Adenylyltransferase 2 (*NMNAT2*), and component of vacuolar ATPase (*ATP6V0E2*). The three upregulated genes were Zinc finger C2HC-type (*ZC2HC1A*), monooxygenase DBH-like 1 (*MOXD1*), and protein tyrosine phosphatase, receptor type, f polypeptide, interacting protein alpha 2 (*PPFIA2*).

Gene Set Enrichment Analysis (GSEA) was conducted using the eVITTA platform to explore further the differential expression results obtained from DESeq2 in Figure 1b. This analysis aimed to identify biologically relevant pathways, gene ontologies, and regulatory networks associated with the differentially expressed genes (DEGs) observed across different ethnic groups in papillary thyroid cancer (PTC). By leveraging well-curated pathway databases, GSEA provided a deeper understanding of the functional significance of these transcriptomic differences and their potential role in tumor progression, immune response, and cellular metabolism. The GSEA approach enabled the identification of enriched pathways that could offer insights into the molecular mechanisms driving racial and ethnic disparities in PTC. Specifically, key signaling pathways related to oncogenesis, immune system regulation, DNA damage repair, cell cycle progression, and metabolic reprogramming were evaluated. The analysis allowed the comparison of pathway activation patterns between ethnic groups.

In addition to pathway enrichment, gene ontology (GO) term analysis was performed to classify DEGs based on their biological functions, cellular localization, and molecular activities. This provided a comprehensive view of how genetic differences may influence various aspects of PTC pathophysiology. Notably, immune-related pathways such as cytokine signaling, antigen presentation, and inflammatory response were differentially enriched among ethnic groups, suggesting variations in tumor–immune interactions. Furthermore, alterations in metabolic and endocrine-related pathways, including thyroid hormone biosynthesis and glucose metabolism, were identified, providing potential explanations for disparities in tumor behavior and treatment response.

Table 1 shows the top 10 differentially expressed (upregulated and downregulated) genes based on more than 2.5-fold higher or lower expression with significant *p*-values (*p* < 0.01). Overall, the GSEA findings underscore the importance of molecular-level investigations in understanding racial and ethnic differences in PTC outcomes.

### 3.2. Gene Expression Profiles: Comparison Between Hispanic and African American Papillary Thyroid Cancer Tissue Samples

The comparison between Hispanic and African American PTC samples revealed 2012 differentially expressed genes (*p* < 0.05). The top 10 differentially expressed genes in White vs. African American PTC samples are shown in the Volcano plot by the adjusted *p*-value of <0.05 (Figure 2a). The two downregulated genes are PIWI-like RNA-mediated gene silencing protein (*PIWIL1*) and Zinc finger BED-type containing 2 (*ZBED2*). The eight upregulated genes are Zinc finger protein (*ZNF700*), general transcription factor IIH subunit 3 (*GTF2H3*), ribosomal protein L17 (*RPL17*), ELL-associated factor 2 (*EAF2*), pyruvate dehydrogenase kinase 1 (*PDK1*), *CD38*, Immunoglobulin heavy variable 4-59 (*GHV4-59*), and Immunoglobulin lambda variable 1-47 (*IGLV1-47*).

Gene Set Enrichment Analysis (GSEA) for Hispanic versus African American PTC samples was conducted using the eVITTA platform to gain deeper insights into the functional significance of the differential gene expression results obtained from DESeq2 (Figure 2b). This analysis aimed to identify enriched biological pathways, gene ontologies, and regulatory networks associated with the observed transcriptomic differences between these two ethnic groups. By leveraging curated gene sets and pathway databases, GSEA helped to determine whether specific biological processes were over-represented among differentially expressed genes.

The analysis focused on pathways related to tumor progression, immune regulation, cellular metabolism, and thyroid-specific signaling, providing a functional context for the molecular differences contributing to potential disparities in PTC pathogenesis and outcomes. TCGA analysis revealed significant upregulation of *PIWIL1* and downregulation of *ZNF700, GTF2H3, ZBED2, RPL17*, and *EAF2* across all three racial groups. *ZNF700* exhibited a significant difference between White and Asian patients (*p* < 0.01). In contrast, PDK1 was significantly downregulated only in African American patients (*p* < 0.01), and its expression was also associated with survival (*p* = 0.02). *CD38* was significantly downregulated in African and Asian Americans (*p* < 0.01) with no correlation with survival. Table 2 shows the top differentially expressed genes (six upregulated and four downregulated) based on more than 2.5-fold higher or lower expression with significant *p*-values (*p* < 0.01).

### 3.3. Gene Expression Profiles: Comparison Between White and African American Papillary Thyroid Cancer Tissue Samples 

The comparison between White and African American PTC samples revealed 1702 differentially expressed genes (*p* < 0.05). The two downregulated genes are Placental Growth Factor (PGF) and Papillary Thyroid Carcinoma Susceptibility Candidate 1 (PTCSC1). The two upregulated genes are the Rapamycin-insensitive companion of mTOR (RICTOR) and Nuclear Pore Complex Interacting Protein Family Member A3 (NPIPA3). The comparison revealed the top 10 differentially expressed genes in the Volcano plot by an adjusted *p*-value of <0.05 (Figure 3a).

Gene Set Enrichment Analysis (GSEA) was conducted. This analysis aimed to identify key biological pathways, gene ontologies, and molecular networks that were significantly enriched among DEGs between these two groups. The eVITTA platform was used to further explore the differential expression results obtained from DESeq2 in Figure 3b.

Table 3 shows the top 10 differentially expressed upregulated and downregulated genes based on more than 2.5-fold higher or lower expression with significant *p*-values (*p* < 0.01). We also showed the probable function of these up- and downregulated genes in cancer. The analysis focused on pathways related to tumor progression, immune response regulation, cellular metabolism, and thyroid hormone signaling.

Notably, African American patients exhibited enrichment in pathways associated with immune suppression, reduced tumor suppressor activity, and altered inflammatory signaling, suggesting potential differences in tumor immune evasion and microenvironment interactions. In contrast, White patients showed enrichment in pathways linked to cell cycle regulation, DNA repair mechanisms, and metabolic processes. TCGA analysis showed PGF, PTCSC1, and RICTOR significantly (*p* < 0.001) downregulated across all racial groups and sample cohorts.

### 3.4. Gene Expression Profiles: Comparison Between White and Hispanic Papillary Thyroid Cancer Tissue Samples

The comparison between White and Hispanic PTC samples revealed 1253 differentially expressed genes (*p* < 0.05). We found 16 downregulated genes: Testis-specific protein Y-encoded 4 (*TSPY4*), Immunoglobulin kappa variable 3-15 (*IGKV3-15*), Immunoglobulin heavy variable 2-26 (*IGHV2-26*), Immunoglobulin heavy variable 3-43 (*IGHV3-43*), Lymphocyte transmembrane adaptor 1 (*LAX1*), X-box binding protein 1 (*XBP1*), Immunoglobulin kappa variable 1D-13 (*IGKV1D-13*), Immunoglobulin heavy variable 3-33 (*IGHV3-33*), Immunoglobulin lambda variable 2-18 (*IGLV2-18*), Thioredoxin domain containing 5 (*TXNDC5*), Immunoglobulin kappa variable 1D-39 (*IGKV1D-39*), Immunoglobulin lambda constant 1 (*IGLC1*), Fc receptor-like 5 (*FCRL5*), Immunoglobulin heavy variable 3-13 (*IGHV3-13*), Immunoglobulin heavy variable 3-20 (*IGHV3-20*), and Immunoglobulin heavy constant mu (*IGHM*). The comparison showed the top 10 differentially expressed genes in the Volcano plot by an adjusted *p*-value of <0.05.

Gene Set Enrichment Analysis (GSEA) was conducted using the eVITTA platform to explore the differential expression results obtained from DESeq2 in Figure 4b. This analysis aimed to identify enriched biological pathways, gene ontologies, and molecular mechanisms underlying the transcriptomic differences between these two ethnic groups. Using curated gene sets and pathway databases, GSEA provided insights into how DEGs contribute to distinct biological processes, potentially influencing PTC pathogenesis, tumor behavior, and patient outcomes.

The analysis focused on pathways involved in tumor progression, immune system modulation, cellular metabolism, and thyroid-specific signaling networks. Hispanic patients exhibited enrichment in pathways related to inflammatory response, oxidative stress regulation, and epithelial-to-mesenchymal transition (EMT). In contrast, White patients demonstrated significant enrichment in pathways associated with cell cycle control, DNA damage repair, and metabolic reprogramming.

Table 4 shows the top 16 differentially expressed downregulated genes based on more than 2.5-fold higher or lower expressions with significant *p*-values (*p* < 0.01).

### 3.5. Correlation of Differentially Expressed Genes from the LLU Cohort to the TCGA Cohort in African Americans, Whites, and Asians

The TCGA cohort includes data from White, African American, and Asian American individuals but does not include Hispanic Americans due to genetic admixture. Gene expression analysis from the TCGA database revealed significant upregulation of *MIR205HG* and *SLC16A9* in normal tissues compared to White, African American, and Asian patient cohorts (*p* < 0.01). Conversely, *KCNQ1*, *ATP6V0E2*, and *PPFIA* were downregulated across these groups. Furthermore, the downregulation of *HYOU1* and *PPFIA2* was significantly correlated with survivability (*p* = 0.03) (Appendix A). Among all analyzed genes, SOD3 exhibited the most significant differential expression between White and African American patients (*p* < 0.001) and *ATP6V0E2* showed significant differential expression between African and Asian patients.

TCGA data analysis showed that *LAX1* and *TXNDC5* were significantly downregulated in White and African American samples, while *XBP1* was significantly downregulated across all three racial groups. Overall, TCGA validation strengthened the robustness of our conclusions by confirming reproducibility across an independent, well-characterized dataset. We performed TCGA data analysis using publicly available UALCAN data analysis portal (https://ualcan.path.uab.edu).

Genes with Significant Upregulation: *MIR205HG* is significantly upregulated in Whites (*p* = 0.000006, n = 332) and in African Americans and Asians (*p* = 0.004, n = 51). However, it does not show a significant association with survival. *SLC16A9* is significantly upregulated in Whites (*p* = 0.0001, n = 27), African Americans (*p* = 0.03, n = 334), and Asians (*p* = 0.02, n = 51), but it has no significant correlation with survival. *PIWIL1* is significantly upregulated in Whites (*p* = 0.02, n = 27), African Americans (*p* = 0.000000000001, n = 334), and Asians (*p* = 0.0009, n = 51). However, its expression does not impact survival outcomes.

Genes with Significant Downregulation: *SOD3* is significantly downregulated in Whites (*p* = 0.000098, n = 27), African Americans (*p* = 0.000000000001, n = 334), and Asians (*p* = 0.000000000001, n = 51). However, it does not have a significant association with survival. *HYOU1* is significantly downregulated in Whites (*p* = 0.00002, n = 27), African Americans (*p* = 0.000000000001, n = 334), and Asians (*p* = 0.000000000001, n = 51). It is also significantly associated with survival (*p* = 0.0039). *KCNQ1* is significantly downregulated across all racial groups, with the most pronounced significance in Whites (*p* = 0.000000000009, n = 27), African Americans (*p* = 0.000000000001, n = 334), and Asians (*p* = 0.000000000001, n = 51). However, it does not correlate with survival. *ATP6V0E2* is significantly downregulated in Whites (*p* = 0.01, n = 27), African Americans (*p* = 0.000003, n = 334), and Asians (*p* = 0.0000002, n = 51), but it is not linked to survival outcomes. *PPFIA2* is significantly downregulated in Whites (*p* = 0.006, n = 27), African Americans (*p* = 0.000001, n = 334), and Asians (*p* = 0.00002, n = 51). It also shows a significant correlation with survival (*p* = 0.036). *ZNF700, GTF2H3, ZBED2, RPL17, EAF2, PDK1, CD38, PGF, PTCSC1, RICTOR, LAX1, XBP1,* and *TXNDC5* are all significantly downregulated in Whites, African Americans, and Asians. However, none of these genes show a significant association with survival.

The analysis of gene expression in African American, White, and Asian populations reveals several significant differences (Appendix A).

Below is a detailed summary of the up- and downregulated genes from the TCGA data analysis, which was matched to the LLU cohort, and their association with survival outcomes as shown in Table 5.

## 4. Discussion

Our study provides critical insights into the molecular disparities in papillary thyroid cancer (PTC) across different ethnic groups by identifying distinct gene expression patterns that may contribute to variations in incidence and outcomes. The differential expression of key genes involved in epithelial-to-mesenchymal transition (EMT), immune response, and oncogenic signaling suggests potential mechanisms underlying ethnic-specific susceptibilities to PTC.

A particularly intriguing finding is the downregulation of miR205-hg in Asian PTC samples. miR205-hg hosts microRNA-205, known for its dual role in cancer, functioning as both a tumor suppressor and an oncogene depending on the cancer type and cellular context. miR-205 has been implicated in EMT and cancer progression [17]. It targets *ZEB1* and *ZEB2*, key regulators of EMT. EMT is a critical process in tumor invasion and metastasis, and its dysregulation may influence PTC progression. MIR205HG modulates the tumor microenvironment by immune cell infiltration and cytokine expression, suppressing the oncogenic pathway. The reduced expression of miR205-hg in Asian patients suggests potential alterations in EMT dynamics, which may partially explain the higher incidence of PTC in this population. Reduced MIR205HG may enhance EMT, contributing to increased incidence and progression of PTC in certain ethnic groups (Asians). MIR205HG expression levels could help predict prognosis in PTC; modulating MIR205HG expression may be a potential strategy for cancer treatment. Functional validation in PTC is needed to confirm its role.

Additionally, superoxide dismutase 3 (SOD3)’s role in cancer is multifaceted, acting as both a tumor suppressor and a modulator of the tumor microenvironment. Its expression levels and genetic variations can influence cancer risk, progression, and response to therapy, underscoring the importance of SOD3 in cancer biology, and its suppression in Asian PTC patients may contribute to heightened susceptibility by deregulating key oncogenic pathways. Downregulation in Asian PTC patients suggests a possible genetic or epigenetic mechanism contributing to increased incidence. SOD3 expression is often downregulated in tumors due to hypermethylation of its CpG sites, a phenomenon observed in gallbladder, liver, prostate, lung, and breast cancers [18]. This epigenetic modification contributes to reduced SOD3 levels, potentially promoting tumor progression. Certain SOD3 gene variants are linked to an increased risk of cancers, such as cervical and prostate cancers. These variants may lead to decreased SOD3 expression, contributing to carcinogenesis [19].

Another notable observation is the downregulation of *PIWIL1* (PIWI-like RNA-mediated gene silencing protein 1) in Hispanic PTC samples. *PIWIL1* is involved in stem cell maintenance and has been linked to tumorigenesis [20]. *PIWIL1* is a member of the *PIWI* subfamily of Argonaute proteins, primarily involved in RNA silencing, transposon suppression, and stem cell maintenance. PIWIL1 is frequently associated with cancer stemness, tumor proliferation, and metastasis in various malignancies. Its reduced expression in Hispanic patients may influence disease progression or treatment responses differently compared to African American patients. This highlights the necessity of considering ethnic-specific molecular mechanisms when tailoring therapeutic strategies for PTC.

Furthermore, compared to African American samples, the downregulation of PGF and PTCSC1 in White PTC samples suggests potential differences in tumor aggressiveness. Placental Growth Factor (PGF) is a member of the vascular endothelial growth factor (VEGF) family and is primarily involved in angiogenesis, inflammation, and tumor progression. PGF is known to play a role in various cancers by promoting tumor vascularization, enhancing metastasis, and modulating the tumor microenvironment. However, its specific function in papillary thyroid cancer (PTC) remains underexplored. PGF is often upregulated in cancers and promotes angiogenesis and metastasis [21]. Lower PGF expression in White patients might correlate with a less aggressive tumor phenotype. Epigenetic regulation (e.g., PGF promoter methylation) may explain its differential expression among racial groups. Differences in tumor vascularization and immune infiltration could also be linked to racial disparities. Similarly, PTCSC1, a long non-coding RNA associated with PTC risk [22], exhibits downregulation in White patients, warranting further investigation into its role in disease susceptibility and progression across ethnic groups.

The differential expression of immune-related genes such as IGKV3-15, IGHV2-26, and IGHV3-43 in White vs. Hispanic comparisons underscores potential variations in immune responses to PTC. These findings may have implications for immunotherapy strategies, as differences in immune gene expression could influence treatment efficacy [22]. Understanding these variations may help refine immunotherapeutic approaches tailored to specific ethnic populations.

Our Gene Set Enrichment Analysis (GSEA) further reinforces these findings, highlighting the enrichment of pathways related to EMT, inflammatory response, and cell cycle regulation. These pathways are crucial to cancer progression and may contribute to the observed disparities in PTC incidence and outcomes [23].

TCGA data analysis showed a significant correlation with survivability with HYOU1, PPFIA2, PDK1, and PGF expression.

Despite these significant findings, our study has limitations. The sample sizes for each ethnic group were relatively small, potentially affecting the generalizability of our results. To mitigate this, we plan to validate our findings using publicly available datasets with comprehensive patient demographic information. Additionally, functional studies are necessary to elucidate the biological significance of the differentially expressed genes and their potential as therapeutic targets. By continuing to investigate ethnic disparities at a molecular level, we aim to contribute to developing personalized interventions that can improve PTC outcomes for diverse patient populations. The analysis identifies several genes that show significant differences based on sample type, race, and survival outcomes in White vs. Asian populations. SOD3, HYOU1, and PPFIA2 show the strongest association with both racial differences and survival, indicating a potential role in racial disparities in cancer outcomes. MIR205HG, SLC16A9, KCNQ1, and ATP6V0E2 are differentially expressed based on race and sample type but do not show a significant survival impact. ZC2HC1A (FAM164A) and MOXD1 do not exhibit significant differences, suggesting they are not key players in racial disparities in this dataset.

When we analyzed and validated our genes with the TCGA cohort, HYOU1 and PPFIA2 were significantly downregulated across racial groups; these genes are associated with survival, indicating their potential prognostic value. MIR205HG, PIWIL1, and SLC16A9 are the only genes that were upregulated in cancer samples across all racial groups, but they do not impact survival. Several other genes, including KCNQ1, SOD3, ATP6V0E2, and RICTOR, were significantly downregulated across racial groups, but they do not show survival significance. This analysis highlights potential gene candidates for further investigation into racial disparities in cancer biology.

## 5. Conclusions

Our study builds upon previous research demonstrating differential protein and miRNA expression between White and Asian (Filipino American) PTC patients. In this study, we extended the analysis to include African American and Hispanic populations. This broader scope enhances our understanding of ethnic disparities in PTC at the molecular level.

In conclusion, our study identifies distinct gene expression patterns in PTC across racial and ethnic groups, shedding light on potential biological contributors to disparities in incidence and outcomes. The differentially expressed genes and pathways uncovered in this analysis provide a foundation for future research into ethnicity-specific biomarkers and targeted therapies. These findings highlight the critical need to integrate racial and ethnic considerations into PTC research and clinical management, advancing personalized and more effective treatment strategies.

## Figures and Tables

**Figure 1 curroncol-32-00315-f001:**
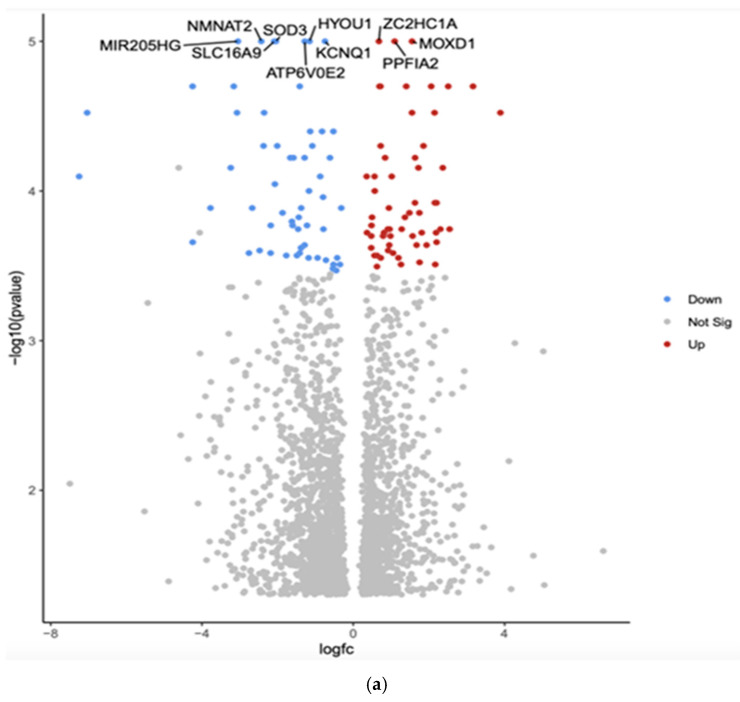
(**a**) Volcano plot of differentially expressed genes in White versus Asian PTC samples. The *x*-axis represents the log2 fold change, and the *y*-axis represents the −log10 adjusted *p*-value. Red dots indicate significantly upregulated genes in Whites compared to Asians, blue dots indicate significantly downregulated genes in Whites compared to Asians, and gray dots represent genes with no significant change. The top 10 differentially expressed genes are labeled. (**b**) Gene Set Enrichment Analysis (GSEA) for Whites versus Asians was conducted using the eVITTA platform to explore the differential expression results obtained from DESeq2 further. The GSEA approach enabled the identification of enriched pathways that could offer insights into the molecular mechanisms driving racial and ethnic disparities in PTC. Specifically, key signaling pathways related to oncogenesis, immune system regulation, DNA damage repair, cell cycle progression, and metabolic reprogramming were evaluated.

**Figure 2 curroncol-32-00315-f002:**
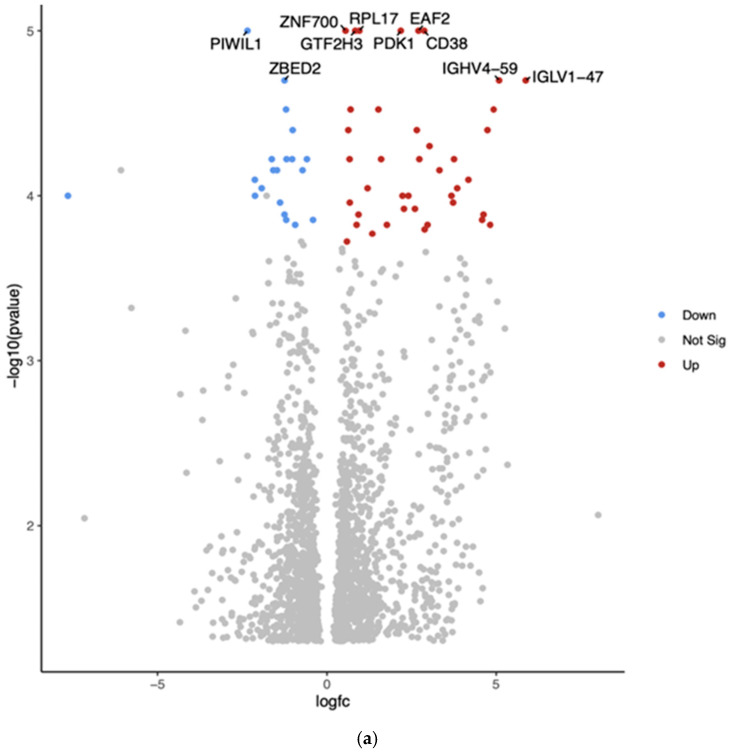
(**a**) Volcano plot of differentially expressed genes in Hispanic versus African American PTC samples. The *x*-axis represents the log2 fold change, and the *y*-axis represents the −log10 adjusted *p*-value. Red dots indicate significantly upregulated genes, blue dots indicate significantly downregulated genes in Hispanics compared to Africans, and gray dots represent genes with no significant change. The top 10 differentially expressed genes are labeled. (**b**) Gene Set Enrichment Analysis (GSEA) for Hispanic versus African American PTC samples was conducted using the eVITTA platform to gain deeper insights into the functional significance of the differential gene expression results obtained from DESeq2. The analysis focused on pathways related to tumor progression, immune regulation, cellular metabolism, and thyroid-specific signaling, providing a functional context for the molecular differences contributing to potential disparities in PTC pathogenesis and outcomes.

**Figure 3 curroncol-32-00315-f003:**
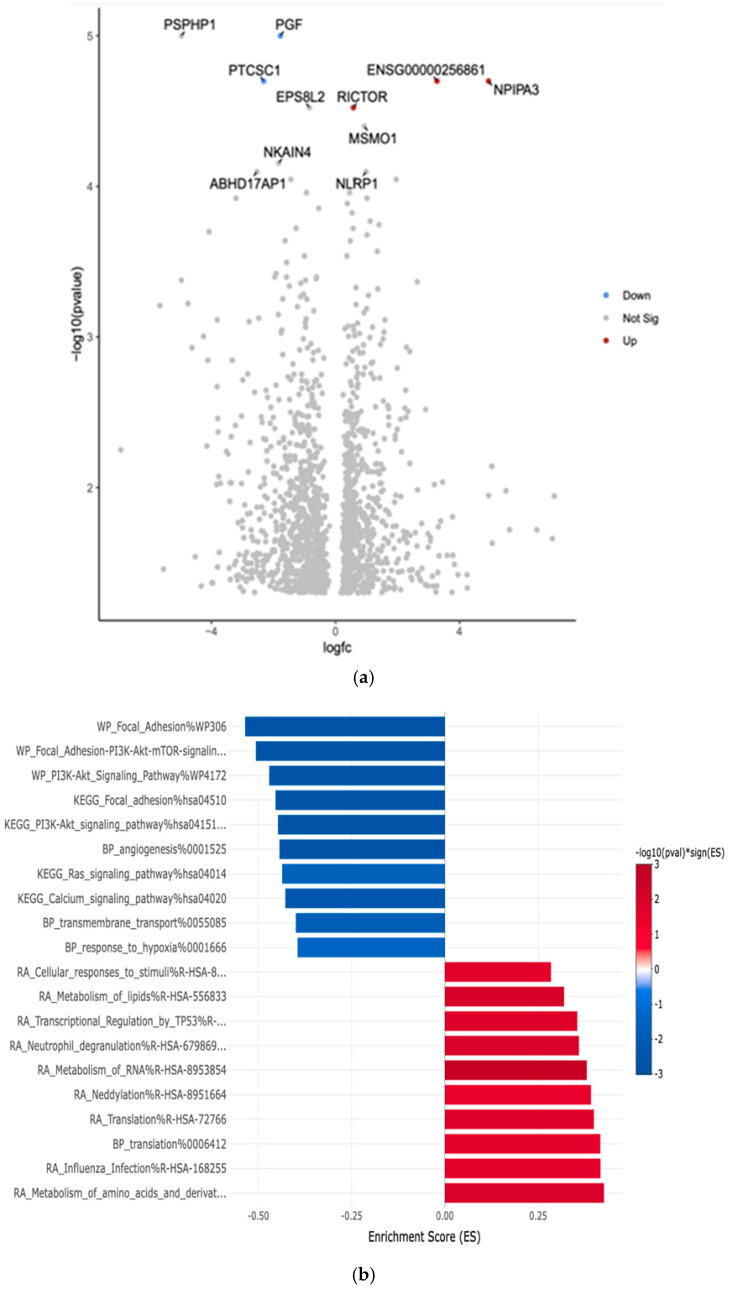
(**a**) Volcano plot of differentially expressed genes in White versus African American PTC samples. The *x*-axis represents the log2 fold change, and the *y*-axis represents the −log10 adjusted *p*-value. Red dots indicate significantly upregulated genes, blue dots indicate significantly downregulated genes in Whites compared to Africans, and gray dots represent genes with no significant change. The top 10 differentially expressed genes are labeled. (**b**) Gene Set Enrichment Analysis (GSEA) for White versus African American PTC samples was conducted using the eVITTA platform to further investigate the differential expression results obtained from DESeq2. By leveraging curated pathway databases, GSEA provided insights into the functional implications of transcriptomic differences, revealing potential mechanisms contributing to racial disparities in PTC.

**Figure 4 curroncol-32-00315-f004:**
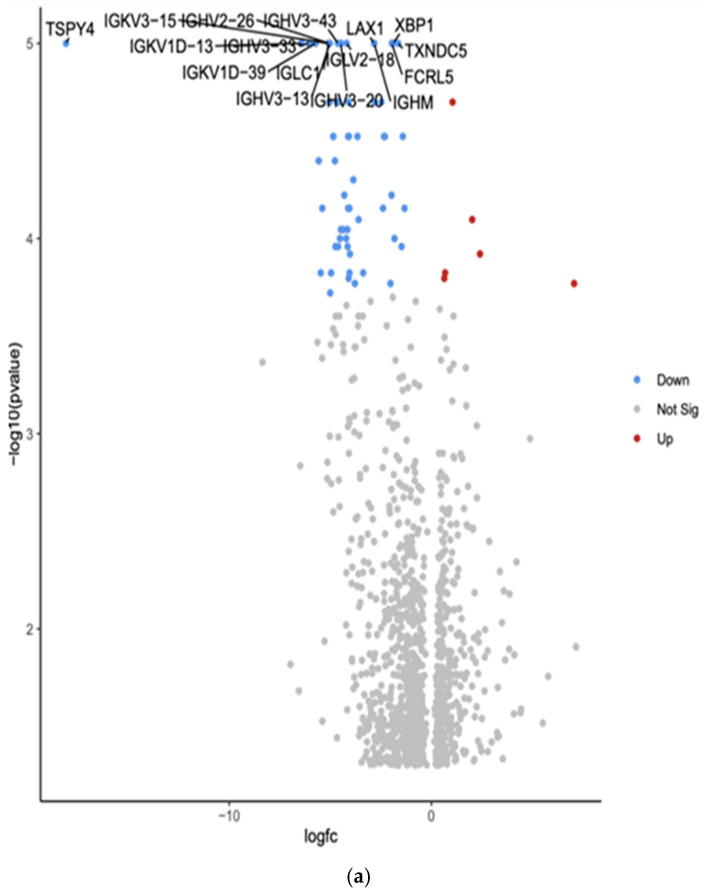
(**a**) Volcano plot of differentially expressed genes in White versus Hispanic PTC samples. The *x*-axis represents the log2 fold change, and the *y*-axis represents the −log10 adjusted *p*-value. Red dots indicate significantly upregulated genes, blue dots indicate significantly downregulated genes in Hispanics compared to Whites, and gray dots represent genes with no significant change. The top 10 differentially expressed genes are labeled. (**b**) Gene Set Enrichment Analysis (GSEA) for White versus Hispanic PTC samples was conducted using the eVITTA platform to further explore the differential expression results obtained from DESeq2. Using curated gene sets and pathway databases, GSEA provided insights into how differentially expressed genes (DEGs) contribute to distinct biological processes, potentially influencing PTC pathogenesis, tumor behavior, and patient outcomes.

**Table 1 curroncol-32-00315-t001:** List of 10 significant genes in White versus Asian PTC samples.

Gene	Fold Change	Function	Relation to Cancer
*MIR205HG*	Downregulated −3.05 (***, *p* = 1 × 10^−5^)	Long non-coding RNA that hosts microRNA-205; involved in epithelial-to-mesenchymal transition (EMT)	Associated with various cancers, including breast and prostate cancer; can act as both a tumor suppressor and an oncogene
*SOD3*	Downregulated −2.05 (*** *p* = 1 × 10^−5^)	SOD3 has anti-oxidative, anti-inflammatory, anti-apoptotic, and growth-promoting characteristics in cardiovascular and cancer models	Tumor suppression, increased expression in a benign thyroid tumor goiter model and gradually downregulated in cell lines that model advanced papillary and anaplastic thyroid cancers
*HYOU1*	Downregulated −1.16 (***, *p* = 1 × 10^−5^)	Hypoxia upregulated protein 1; involved in protein folding and secretion under stress conditions	Overexpression is associated with poor prognosis in some cancers; promotes tumor growth and metastasis
*SLC16A9*	Downregulated −2.09 (***, *p* = 1 × 10^−5^)	Solute carrier family 16 member 9; monocarboxylate transporter	Involved in metabolic reprogramming in cancer cells; may contribute to tumor growth
*KCNQ1*	Downregulated −0.75 (***, *p* = 1 × 10^−5^)	Potassium voltage-gated channel; involved in cardiac action potential	Acts as a tumor suppressor in some cancers; its downregulation may promote cancer progression
*ATP6V0E2*	Downregulated −1.28 (***, *p* = 1 × 10^−5^)	Component of vacuolar ATPase; involved in cellular pH regulation	May contribute to cancer cell survival and metastasis by regulating pH
*ZC2HC1A*	Upregulated 0.68 (***, *p* = 1 × 10^−5^)	Zinc finger C2HC-type containing 1A; function not well characterized	Limited information on its role in cancer
*MOXD1*	Upregulated 1.55 (***, *p* = 1 × 10^−5^)	Monooxygenase DBH-like 1; involved in catecholamine metabolism	Limited information on its role in cancer
*PPFIA2*	Upregulated 1.09 (***, *p* = 1 × 10^−5^)	Protein tyrosine phosphatase, receptor type, f polypeptide, interacting protein alpha 2; involved in cell adhesion	May play a role in cancer cell invasion and metastasis
*NMNAT2*	Downregulated −2.43 (*** *p* = 1 × 10^−5^)	Nicotinamide Mononucleotide Adenylyltransferase 2 (NMNAT2) is an enzyme that helps maintain axons and protects neurons; it is encoded by the NMNAT2 gene in humans	NMNAT2 exerts a cancer-promoting role in solid tumors, including colorectal cancer, lung cancer, ovarian cancer, and glioma, and is closely related to tumor occurrence and development

*p* < 0.0001 (***).

**Table 2 curroncol-32-00315-t002:** Comparison of the top 10 differentially expressed genes in Hispanic vs. African American PTC samples.

Gene	Fold Change	Function	Relation to Cancer
*PIWIL1*	Downregulated −2.35 (***, *p* = 1.00 × 10^−5^)	PIWI-like RNA-mediated gene-silencing protein	Potential tumor suppressor
*ZNF700*	Upregulated 0.55 (***, *p* = 1.00 × 10^−5^)	Zinc finger protein	Limited information on its role in cancer
*GTF2H3*	Upregulated 0.83 (***, *p* = 1.00 × 10^−5^)	General transcription factor IIH subunit 3	May play a role in DNA repair and genomic stability
*ZBED2*	Downregulated −1.25 (***, *p* = 2.00 × 10^−5^)	Zinc finger BED-type containing 2	Limited information on its role in cancer
*RPL17*	Upregulated 0.95 (***, *p* = 1.00 × 10^−5^)	Ribosomal protein L17	Often dysregulated in cancer
*EAF2*	Upregulated 2.70 (***, *p* = 1.00 × 10^−5^)	ELL-associated factor 2	Potential tumor suppressor in some cancers
*PDK1*	Upregulated 2.18 (***, *p* = 1.00 × 10^−5^)	Pyruvate dehydrogenase kinase 1	Often upregulated in cancer; contributes to metabolic reprogramming
*CD38*	Upregulated 2.87 (***, *p* = 1.00 × 10^−5^)	Cyclic ADP-ribose hydrolase	Plays various roles in different cancers
*IGHV4-59*	Upregulated 5.09 (***, *p* = 2.00 × 10^−5^)	Immunoglobulin heavy variable 4-59	May be involved in immune responses to cancer
*IGLV1-47*	Upregulated 5.87 (***, *p* = 2.00 × 10^−5^)	Immunoglobulin lambda variable 1-47	May be involved in immune responses to cancer

(*** *p* < 0.0001).

**Table 3 curroncol-32-00315-t003:** Comparison of the top differentially expressed genes in White vs. African American PTC samples.

Gene	Fold Change	Function	Relation to Cancer
*PGF*	Downregulated −1.78 (***, *p* = 1.00 × 10^−5^)	Placental Growth Factor	Often upregulated in various cancers; promotes tumor angiogenesis and metastasis
*PTCSC1*	Downregulated −2.32 (***, *p* = 2.00 × 10^−5^)	Papillary Thyroid Carcinoma Susceptibility Candidate 1	Associated with increased risk of papillary thyroid cancer; potential tumor suppressor
*RICTOR*	Upregulated 0.58 (***, *p* = 3.00 × 10^−5^)	Rapamycin-insensitive companion of mTOR	Often dysregulated in cancer; involved in cell growth, survival, and metabolism
*NPIPA3*	Upregulated 4.93 (***, *p* = 2.00 × 10^−5^)	Nuclear Pore Complex Interacting Protein Family Member A3	Limited information on its direct role in cancer

(*** *p* < 0.0001).

**Table 4 curroncol-32-00315-t004:** Comparison of the top differentially expressed genes in White vs. Hispanic PTC samples.

Gene	Fold Change	Function	Relation to Cancer
*TSPY4*	Downregulated −18.07 (***, *p* = 1.00 × 10^−5^)	Testis-specific protein Y-encoded 4; involved in cell proliferation and differentiation	May be involved in testicular cancer
*IGKV3-15*	Downregulated −5.09 (***, *p* = 1.00 × 10^−5^)	Immunoglobulin kappa variable 3-15; part of antibody light chains	May be involved in immune responses to cancer
*IGHV3-43*	Downregulated −4.63 (***, *p* = 1.00 × 10^−5^)	Immunoglobulin heavy variable 3-43; part of antibody heavy chains	May be involved in immune responses to cancer
*LAX1*	Downregulated −2.84 (***, *p* = 1.00 × 10^−5^)	Lymphocyte transmembrane adaptor 1; involved in T cell signaling	May play a role in immune responses to cancer; potential tumor suppressor in some cancers
*XBP1*	Downregulated −1.89 (***, *p* = 1.00 × 10^−5^)	X-box binding protein 1; transcription factor involved in unfolded protein response	Often dysregulated in cancer; plays a role in tumor cell survival and drug resistance
*IGKV1D-13*	Downregulated −4.54 (***, *p* = 1.00 × 10^−5^)	Immunoglobulin kappa variable 1D-13; part of antibody light chains	May be involved in immune responses to cancer
*IGHV3-33*	Downregulated −6.04 (***, *p* = 1.00 × 10^−5^)	Immunoglobulin heavy variable 3-33; part of antibody heavy chains	May be involved in immune responses to cancer
*IGLV2-18*	Downregulated −4.19 (***, *p* = 1.00 × 10^−5^)	Immunoglobulin lambda variable 2-18; part of antibody light chains	May be involved in immune responses to cancer
*TXNDC5*	Downregulated −1.65 (***, *p* = 1.00 × 10^−5^)	Thioredoxin domain containing 5; involved in protein folding and redox regulation	Often upregulated in various cancers, may promote tumor growth and metastasis
*IGKV1D-39*	Downregulated −5.73 (***, *p* = 1.00 × 10^−5^)	Immunoglobulin kappa variable 1D-39; part of antibody light chains	May be involved in immune responses to cancer
*IGLC1*	Downregulated −5.09 (***, *p* = 1.00 × 10^−5^)	Immunoglobulin lambda constant 1; part of antibody light chains	May be involved in immune responses to cancer
*FCRL5*	Downregulated −1.96 (***, *p* = 1.00 × 10^−5^)	Fc receptor-like 5; involved in B cell activation and differentiation	May play a role in B cell malignancies; potential therapeutic target in some cancers
*IGHV3-13*	Downregulated −5.02 (***, *p* = 1.00 × 10^−5^)	Immunoglobulin heavy variable 3-13; part of antibody heavy chains	May be involved in immune responses to cancer
*IGHV3-20*	Downregulated −4.48 (***, *p* = 1.00 × 10^−5^)	Immunoglobulin heavy variable 3-20; part of antibody heavy chains	May be involved in immune responses to cancer
*IGHM*	Downregulated −2.83 (***, *p* = 1.00 × 10^−5^)	Immunoglobulin heavy constant mu; part of IgM antibodies	May be involved in immune responses to cancer; altered expression in some B cell malignancies

(*** *p* < 0.0001).

**Table 5 curroncol-32-00315-t005:** TCGA analysis in African Americans, Whites, and Asians.

RNA-Seq	African American (n = 27)	White (n = 334)	Asian (n = 51)	Survival
*MIR205HG*	-	⬆ *** *p* = 0.000006	⬆ ** *p* = 0.004	No Significance
*SOD3*	⬇ *** *p* = 0.000098	⬇ *** *p* = 0.000000000001	⬇ *** *p* = 0.000000000001	No Significance
*HYOU1*	⬇ *** *p* = 0.00002	⬇ *** *p* = 0.000000000001	⬇ *** *p* = 0.000000000001	*** p =* 0.0039
*SLC16A9*	⬆ *** p = 0.0001	⬆ * *p* = 0.03	⬆ * *p* = 0.02	No Significance
*KCNQ1*	⬇ *** *p* = 0.000000000009	⬇ *** *p* = 0.000000000001	⬇ *** *p* = 0.000000000001	No Significance
*ATP6V0E2*	⬇ * *p* = 0.01	⬇ *** *p* = 0.000003	⬇ *** *p* = 0.0000002	No Significance
*PPFIA2*	⬇ ** *p* = 0.006	⬇ *** *p* = 0.000001	⬇ *** *p* = 0.00002	** p =* 0.036
*PIWIL1*	⬆ * *p* = 0.02	⬆ *** *p* = 0.000000000001	⬆ *** *p* = 0.0009	No Significance
*ZNF700*	⬇ ** *p* = 0.009	⬇ *** *p* = 0.000001	⬇ * *p* = 0.03	No Significance
*GTF2H3*	⬇ * *p* = 0.04	⬇ *** *p* = 0.000002	⬇ *** *p* = 0.00003	No Significance
*ZBED2*	⬇ ** *p* = 0.003	⬇ *** *p* = 0.0003	⬇ * *p* = 0.01	No Significance
*RPL17*	⬇ ** *p* = 0.001	⬇ *** *p* = 0.00000001	⬇ *** *p* = 0.00003	No Significance
*EAF2*	⬇ ** *p* = 0.0009	⬇ ** *p* = 0.004	⬇ * *p* = 0.01	No Significance
*PDK1*	⬇ * *p* = 0.03	-	-	** p =* 0.026
*CD38*	⬇ * *p* = 0.01	⬇ * *p* = 0.01	-	No Significance
*PGF*	⬇ ** *p* = 0.003	⬇ *** *p* = 0.0001	⬇ *** *p* = 0.0000008	No Significance
*PTCSC1*	⬇ * *p* = 0.01	⬇ *** *p* = 0.000001	⬇ ** *p* = 0.009	No Significance
*RICTOR*	⬇ *** *p* = 0.000003	⬇ *** *p* = 0.000000000001	⬇ *** *p* = 0.000000001	No Significance
*LAX1*	⬇ * *p* = 0.01	⬇ * *p* = 0.04	-	No Significance
*XBP1*	⬇ *** *p* = 0.000000000009	⬇ *** *p* = 0.000000000001	⬇ *** *p* = 0.000000000005	No Significance
*TXNDC5*	⬇ ** *p* = 0.005	⬇ * *p* = 0.02	-	No Significance

n, number; ⬇, downregulated; ⬆, upregulated; -, no change; (* *p* < 0.05; ** *p* < 0.01; and *** *p* < 0.0001).

## Data Availability

We will provide the raw data that were used to conduct the analyses upon request. Sequence data will be submitted to the public database. Raw data for this study were generated at CARIS Life Sciences. Derived data supporting the findings of this study are available from the corresponding author upon request.

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
