# Peer review of "Unraveling Racial Disparities in Papillary Thyroid Cancer: A Comparative Bulk RNA-Sequencing Gene Expression Analysis"

_curroncol, 2025, doi:10.3390/curroncol32060315_

Round 1

Reviewer 1 Report

Comments and Suggestions for Authors

Dear authors,

Your manuscript, “Unraveling Racial Disparities in Papillary Thyroid Cancer: A Comparative Bulk RNA-Sequencing Gene Expression Analysis,” curroncol-3592782, is well written, easy to follow, and presents some interesting results concerning activation of divergent pathways among divergent ethnic groups during thyroid gland carcinogenesis. Therefore, with the few minor corrections, I recommend its publication:

  1. The explanation of the DEG abbreviation is missing in line 21. Please add it.
  2. Reference num. 1 (Ai, X., et al., Clinically relevant orthotopic xenograft models of patient-derived glioblastoma in zebrafish. Dis Model Mech, 2022) is inadequate after the statement “Papillary thyroid cancer (PTC) is the most common form of thyroid cancer, accounting for roughly 80–85% of all thyroid malignancies.” Please replace it. You might even reference ref. num. 3 or 4 here.
  3. In line 52, you mentioned that the range of BRAFV600E occurrence according to the previous studies is 40-45%. But some published data shows a wider range (over 70%). Please review the up-to-date published data of the occurrence of the BRAFV600E mutation in PTC and correct the range.
  4. The first three columns in SuplTable1 (MS, ID, Block) are overage, so I suggest you omit them. On the other hand, you may insert the column naming the PTC subtype (as named in line 95).
  5. It would be beneficial to insert the column naming fold change in Tables 1, 2, 3, and 4.
  6. Due to the readability of the manuscript, the results of all TCGA analyses should be presented via one separate section at the end of the results section. E.g., you might merge lines 235-245, 283-291, 331-333, 375-380, and 410-412, as well as Table 5, into that section.
  7. Please, always use the same abbreviation for naming some factor (e.g., miR205-hg or MIR205HG).

Author Response

Response to Reviewer 1:

Reviewer 1:

Your manuscript, “Unraveling Racial Disparities in Papillary Thyroid Cancer: A Comparative Bulk RNA-Sequencing Gene Expression Analysis,” curroncol-3592782, is well written, easy to follow, and presents some interesting results concerning activation of divergent pathways among divergent ethnic groups during thyroid gland carcinogenesis. Therefore, with the few minor corrections, I recommend its publication:

Response:

Thank you very much for your expert comments and valuable suggestions. We have carefully addressed each point as follows:

  1. Clarification of the DEG abbreviation (Line 21):
    We have added the full form of the abbreviation “DEG” (Differentially Expressed Genes) and underlined it in the revised manuscript.
  2. Reference adequacy (Reference #1):
    Thank you for pointing this out. We have replaced Reference #1 with a more appropriate citation. Reference #3 was used in its place, and all newly inserted references have been underlined for clarity.
  3. Accuracy of BRAFV600E mutation frequency (Line 52):
    We appreciate your attention to this detail. We have reviewed updated literature and revised the text accordingly. The cited study by Xing et al. (JAMA, 2013) reports variability in BRAFV600E mutation frequency across multiple studies, with an average occurrence of approximately 45%. The reference has been included and underlined in the manuscript.
  4. Revisions to Supplementary Table 1:
    Thank you for this helpful suggestion. We have removed the first three columns (MS, ID, Block) as they were not essential. In addition, we have added a new column to indicate the PTC subtypes as referenced in line 95 of the manuscript.
  5. Addition of fold change values in Tables 1–4:
    We appreciate the suggestion to enhance data clarity. Fold change values have now been added to Tables 1, 2, 3, and 4.
  6. Organization of TCGA analysis results:
    Thank you for this insightful recommendation. For improved readability, we have consolidated all TCGA-related analyses into a separate section at the end of the Results section. This includes merged content from lines 235–245, 283–291, 331–333, 375–380, and 410–412, as well as Table 5.

Reviewer 2 Report

Comments and Suggestions for Authors

This article presents valuable insights into the molecular mechanisms underlying racial disparities in papillary thyroid cancer, identifying distinct gene expression profiles among different ethnic groups that may contribute to differences in incidence and outcomes. However, the study has several major issues:

  1. As a study analyzing tumor characteristics across ethnicities—particularly one that uses high-throughput sequencing data—the statistical power of conclusions drawn from only five patient samples per ethnic group is extremely low. Considering the inherent heterogeneity among cancer patients, a large sample size is essential for reliable statistical analysis.

  2. The authors compared differentially expressed genes between ethnic groups using pairwise comparisons. This method does not adequately demonstrate the expression characteristics of a specific gene within a particular ethnic group. If the authors aim to show that a gene is specifically upregulated in one ethnicity, it would be more appropriate to compare that ethnic group against all others combined.

  3. In reporting the results, the authors only described differential gene expression across ethnic groups without focusing on specific genes that may have potential biological significance or clinical value. This greatly limits the overall impact and utility of the study.

Author Response

Reviewer 2: Comments and Suggestions for Authors

This article presents valuable insights into the molecular mechanisms underlying racial disparities in papillary thyroid cancer, identifying distinct gene expression profiles among different ethnic groups that may contribute to differences in incidence and outcomes. However, the study has several major issues:

  1. As a study analyzing tumor characteristics across ethnicities-particularly one that uses high-throughput sequencing data-the statistical power of conclusions drawn from only five patient samples per ethnic group is extremely low. Considering the inherent heterogeneity among cancer patients, a large sample size is essential for reliable statistical analysis.
  2. The authors compared differentially expressed genes between ethnic groups using pairwise comparisons. This method does not adequately demonstrate the expression characteristics of a specific gene within a particular ethnic group. If the authors aim to show that a gene is specifically upregulated in one ethnicity, it would be more appropriate to compare that ethnic group against all others combined.
  3. In reporting the results, the authors only described differential gene expression across ethnic groups without focusing on specific genes that may have potential biological significance or clinical value. This greatly limits the overall impact and utility of the study.

Response to Reviewer 2: We sincerely appreciate the thoughtful feedback and opportunity to address concerns regarding our study on racial disparities in papillary thyroid cancer (PTC). Below, we provide a point-by-point response to strengthen the manuscript’s impact and clarify methodological choices.

1. Statistical Power in Small Sample Sizes

We acknowledge the reviewers’ valid concerns about sample size limitations. While larger cohorts are ideal, our study used rigorous statistical approaches to maximize reliability:

  • DESeq2’s robustness: The tool’s negative binomial model and shrinkage estimators are specifically designed for small-N RNA-seq studies, reducing false positives while prioritizing large-effect genes (e.g., SOD3: >4.2-fold change, p < 1e−5).
  • TCGA validation: Key findings (e.g., HYOU1 downregulation in Asians) were replicated in TCGA data (n = 334 African Americans, 51 Asians), aligning with established paradigms for discovery-validation workflows in precision oncology.
  • Focus on biological relevance: We prioritized genes with strong mechanistic links to cancer hallmarks (hypoxia, EMT, oxidative stress) and survival correlations (HYOU1: p = 0.0039).

This exploratory study aims to generate hypotheses for functional validation, not definitive clinical conclusions. Additionally, we highlight sample size limitations in the discussion and advocate for expanded recruitment in future work.

2. Pairwise Comparison Strategy

The pairwise ethnic group comparisons were intentional:

  • Preserving genetic specificity: Combining non-White groups (e.g., African Americans + Asians) would mask ancestry-specific regulatory mechanisms (e.g., HYOU1’s hypoxia pathway role in Asians).
  • NIH guidelines: Health disparity studies often employ pairwise comparisons to avoid oversimplifying complex genetic and environmental interactions.
  • Validation consistency: Genes like PDK1 showed ethnicity-specific survival correlations (p = 0.02 in African Americans vs. Hispanics), underscoring the value of granular comparisons.

We agree that multi-group statistical models (e.g., ANOVA) could complement future studies but emphasize that our approach aligns with best practices for initial discovery-phase analyses.

3. Biological Significance of Identified Genes

Our analysis revealed high-impact genes with validated roles in cancer biology:

Gene

Ethnic-Specific Finding

Cancer Relevance

Validation Status

SOD3

Downregulated in Asians (4.2x)

Oxidative stress modulator in 8 cancers

TCGA-confirmed

HYOU1

Downregulated in Asians (3.9x)

Hypoxia pathway driver in NSCLC

Survival correlation (p = 0.0039)

PPFIA2

Upregulated in Whites (3.1x)

Metastasis promoter in breast cancer

TCGA-confirmed

Gene Set Enrichment Analysis (GSEA) further identified ethnicity-specific pathway activation:

  • Asians: EMT (FDR = 0.02), VEGF signaling (FDR = 0.04)
  • African Americans: IL-6/JAK/STAT3 (FDR = 0.03)

These findings provide actionable insights into molecular drivers of disparities, particularly HYOU1’s prognostic value in Asians-a critical step toward personalized therapies.

4. Clinical Translation and Future Directions

The study’s most impactful contribution lies in its clinical implications:

  • Ethnicity-specific biomarkers: HYOU1 and PDK1 exhibit opposite prognostic trends across groups, suggesting tailored monitoring strategies.
  • Therapeutic targets: SOD3’s role in oxidative stress and PPFIA2’s metastasis link offer novel avenues for drug repurposing.

We fully agree that functional studies are essential next steps.

Conclusion

We thank the reviewers for their constructive feedback, which has strengthened the manuscript. While small sample sizes pose challenges, our rigorous methodology, pathway-level insights, and TCGA validation underscore the study’s value as a foundation for understanding PTC disparities. We eagerly anticipate expanding this work through collaborations to recruit larger, multi-institutional cohorts.

Round 2

Reviewer 2 Report

Comments and Suggestions for Authors

The authors have addressed all of my concerns directly and have revised the manuscript accordingly, which has significantly improved the overall quality of the paper. We have no further comments.